# Centrioles are amplified in cycling progenitors of olfactory sensory neurons

Kaitlin Ching[1], Tim Stearns[1,2]*

**1** Department of Biology, Stanford University, Stanford, California, United States of America, **2** Department of Genetics, Stanford University School of Medicine, Stanford, California, United States of America

* stearns@stanford.edu

## Abstract

Olfaction in most animals is mediated by neurons bearing cilia that are accessible to the environment. Olfactory sensory neurons (OSNs) in chordates usually have multiple cilia, each with a centriole at its base. OSNs differentiate from stem cells in the olfactory epithelium, and how the epithelium generates cells with many centrioles is not yet understood. We show that centrioles are amplified via centriole rosette formation in both embryonic development and turnover of the olfactory epithelium in adult mice, and rosette-bearing cells often have free centrioles in addition. Cells with amplified centrioles can go on to divide, with centrioles clustered at each pole. Additionally, we found that centrioles are amplified in immediate neuronal precursors (INPs) concomitant with elevation of mRNA for polo-like kinase 4 (Plk4) and SCL/Tal1-interrupting locus gene (Stil), key regulators of centriole duplication. These results support a model in which centriole amplification occurs during a transient state characterized by elevated Plk4 and Stil in early INP cells. These cells then go on to divide at least once to become OSNs, demonstrating that cell division with amplified centrioles, known to be tolerated in disease states, can occur as part of a normal developmental program.

## Introduction

Olfaction, the primary way that animals sense their chemical environment, begins in olfactory sensory neurons (OSNs). In many chordates, each OSN has multiple cilia, which protrude from the end of a dendrite at the apical surface of the olfactory epithelium. At the apical surface, odorants contact receptors on the surface of cilia, initiating a signaling event in the OSN. At the base of each cilium, a centriole organizes the structure (**Fig 1A**). Cilia are necessary for olfaction, as are the centrioles that organize their microtubule structures [1,2]. To have multiple cilia, each OSN must have multiple centrioles, raising the question: How are these many centrioles made?

The centriole number in OSNs lies between that of two well-studied cases, the centriole pair present in most animal cells and the highly amplified centrioles of multiciliated epithelial cells. Many cell types, and most cycling cells, have exactly two centrioles, with the older of the two often serving as a basal body for a primary cilium (**S1A Fig**). This older centriole is often

**Data Availability Statement:** All relevant data are within the paper and its Supporting Information files.

**Funding:** CMB Training Grant from the National Institutes of Health under award number T32GM007276 (KC), by the National Science

Foundation Graduate Research Fellowship under Grant No. D–G16E5 6518 (KC), and by the National Institutes of Health under award number 1R35GM130286 (TS). Stanford Cell Sciences Imaging Facility supported by National Center for Research Resources under award 1S10RR026780. The funders had no role in study design, data collection and analysis, decision to publish, or preparation of the manuscript.

**Competing interests:** The authors have declared that no competing interests exist.

**Abbreviations:** Arl13b-mCherry, ADP-ribosylation-factor-like GTPase 13b conjugated to mCherry; Cep152, centrosomal protein 152; E, embryonic; eGFP, enhanced green fluorescent protein; GBC, globose basal cell; INP, immediate neuronal precursor; NeuroD1, neuronal differentiation 1; OSN, olfactory sensory neuron; Plk4, polo-like kinase 4; phospho-H3, phosphorylated histone 3; PCNA, proliferating cell nuclear antigen; Rrm2, ribonucleotide reductase molecule 2; scRNAseq, single-cell RNA sequencing; Stil, SCL/Tal1 interrupting locus gene; TEM, transmission electron microscopy.

referred to as the mother centriole and the newer centriole as the daughter centriole. The daughter centriole forms orthogonally to the mother centriole in G1/S phase of the cell cycle and is engaged to the mother until mitosis. Upon passage through mitosis, it becomes disengaged, and in the ensuing cell cycle, it acts as a mother centriole upon which another new daughter centriole can form [3]. In contrast, multiciliated epithelial cells have as many as several hundred centrioles, each serving as a basal body for a motile cilium. In this state, cells exit the cell cycle and initiate a transcriptional program that facilitates this centriole amplification [4–7]. Centriole amplification in multiciliated epithelial cells occurs by two means: (1) centriole growth from deuterosomes, structures that are specific to multiciliated epithelial cells, and (2) by growth of multiple daughter centrioles from each mother centriole, forming rosettes [8]. Centriole rosettes are thought to contribute only a small percentage of the total number of centrioles in multiciliated epithelial cells, although rosette amplification is reported to be sufficient in the absence of deuterosomes [9,10]. Cycling cells can also be induced to form centriole rosettes by overexpression of Polo-like kinase 4 (Plk4), a kinase required for centriole duplication, or by overexpression of certain other centriole duplication proteins [11–15].

Interestingly, centrioles in the olfactory epithelium were previously described to be arranged in a rosette-like array in some cells [16]. However, these centriole rosettes have not been investigated in the context of the cell cycle or differentiation. Our work builds upon this early observation by describing a role for rosettes in centriole amplification, highlighting their formation in cycling progenitor cells, and identifying developmental timing of centriole amplification.

## Results

To better define the range of centriole numbers in our preparations, we counted centrioles in OSNs from mice expressing centrin2 conjugated to enhanced green fluorescent protein (eGFP-centrin2), a marker of the centriole [17]. In nasal septa from adult mice, OSNs had an average of 15.7 centrioles per cell, with wide variation around the mean (6 to 37 centrioles/cell, SD 6.15 centrioles) but no apparent trend across the anterior–posterior axis. This number of centrioles is similar to previous reports of cilium and centriole number in OSNs [2,18].

We next considered the potential means by which cells amplify centriole number during differentiation from stem cells to OSNs. Centrioles in the olfactory epithelium were previously described to be arranged in a rosette-like array in some cells [16]. To assess the presence and role of centriole rosettes in the olfactory epithelium, we visualized centrioles by transmission electron microscopy (TEM) and fluorescence microscopy in both adult and embryonic tissue. First, ultrathin sections were made from dissected olfactory turbinates taken from adult mice and examined by TEM. We observed dendritic knob structures with multiple centrioles, typical of OSNs (**Fig 1A**), as well as horizontal basal cells with centriole pairs and primary cilia (**S1A Fig**). Near the basal lamina, where OSN progenitor cells are typically found, we found cells with centriole rosettes (**Fig 1B**). Next, we determined whether centriole rosettes were present in cryosections of embryonic (E12.5) olfactory epithelia from mice expressing eGFP-centrin2 (**Fig 1C**, **S1B Fig**). Rosettes were apparent as clusters of eGFP-centrin2 foci with the expected dimensions. Note that the cell shown in the inset has two rosettes, consistent with rosette formation on both preexisting centrioles. In addition, we found that rosette-bearing cells were positive for the neuronal marker β-tubulin III, confirming that these cells were committed to a neuronal cell fate (**S1C Fig**). Our results suggest that centrioles are amplified by rosettes in both adult and embryonic olfactory epithelium.

By observing olfactory epithelia of adult mice by TEM and embryonic mice by fluorescence microscopy, we also found cells that had free centrioles in addition to two rosettes. In olfactory

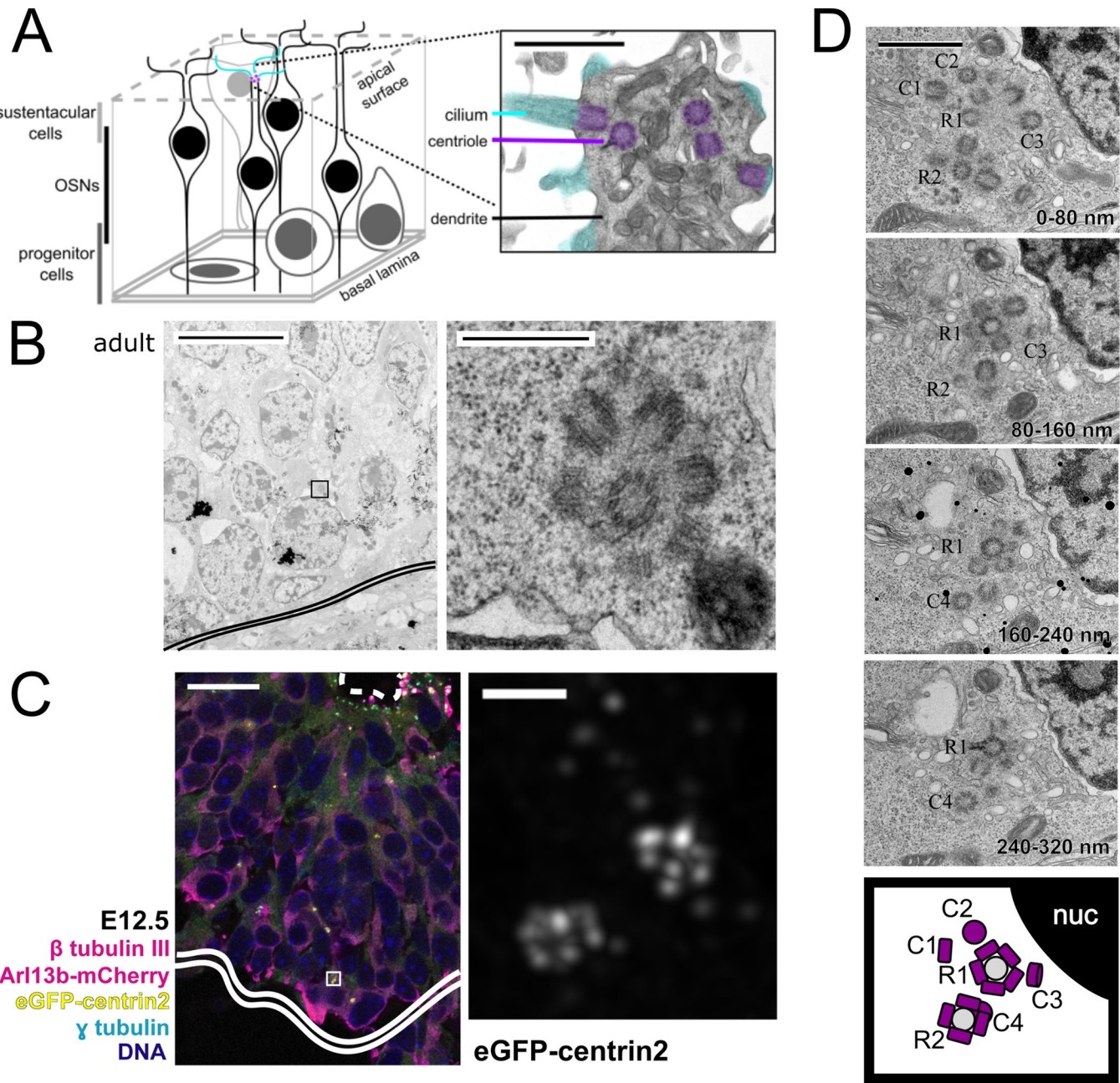

**Fig 1. Location of rosette structures and free centrioles in adult and embryonic mouse olfactory epithelium.** (A) Schematic of the olfactory epithelium. The inset shows an OSN dendrite from adult mouse imaged by TEM with pseudocolored cilia and centrioles. Inset scale bar = 1 μm. (B) TEM image of wild-type adult mouse olfactory epithelium. Double solid line marks the basal lamina. Box shows the approximate location of the inset in the panel to the right. The panel on the right shows an inset of a centriole rosette in cross section, near the basal lamina. Scale bar = 10 μm. Inset scale bar = 0.5 μm. (C) Fluorescence image of embryonic olfactory epithelium at E12.5 in mice expressing eGFP-centrin2 to mark centrioles, as well as Arl13b-mCherry to mark cilia. The maximum projection inset shows a deconvolved image of 2 rosette-like centriole clusters from a cell positive for β tubulin III, near the basal lamina. Dashed line marks the apical surface of the olfactory epithelium. Double solid line marks the basal lamina. Box shows the location and orientation of the inset. Scale bar = 20 μm. Inset scale bar = 1 μm. (D) TEM images of wild-type adult mouse olfactory epithelium in serial sections. The images show 2 centriole rosettes, R1 and R2, and free centrioles, C1-4. Section sequence is indicated in the bottom right of each panel. The bottom panel summarizes the locations of centrioles in all 4 panels, where new centrioles are shown in purple and mother centrioles are shown in gray. Scale bar = 1 μm. See S1 Fig for additional details. Arl13b-mCherry, ADP-ribosylation-factor-like GTPase 13b conjugated to mCherry; eGFP, enhanced green fluorescent protein; nuc, nucleus; OSN, olfactory sensory neuron; TEM, transmission electron microscopy.

epithelia from adult mice, we found cells with two rosettes, one per parent centriole, as well as free centrioles by TEM (**Fig 1D**, **S1D1–S1D4 Fig**). Similarly, puncta of eGFP-centrin2 were observed near rosettes in embryonic olfactory epithelia by fluorescence microscopy (see **Fig 1C**, **S1B Fig**). Whether free centrioles formed by detaching from a centriole rosette or free of a parental centriole (i.e., de novo) requires further investigation.

Next, we next asked whether centriole amplification can occur in cycling cells or only in nondividing differentiated cells, using fluorescence microscopy in adult and embryonic olfactory epithelium. We used stage-specific markers to assess centriole amplification in cells in different stages of the cell cycle. In the olfactory epithelium of wild-type adult mice, some cells with nuclear proliferating cell nuclear antigen (PCNA), a marker for S phase, had centriole rosettes (**S2A and S2A' Fig**). To determine whether cells which amplify centrioles in S phase proceed through mitosis, we probed for phosphorylated histone 3 (phospho-H3) and confirmed the mitotic state by DAPI staining. Previous studies have shown that phosphorylation at serine 10 of the histone subunit H3 gradually increases to become highest at metaphase before the signal decreases and relocalizes away from chromatin during anaphase [19,20]. In the olfactory epithelium of adult mice expressing eGFP-centrin2, many mitotic cells with condensed chromatin and high phospho-H3 had clusters of centrioles (**Fig 2A, 2A'** **and S2B Fig**, **S2C Fig**). When mitotic spindles were observed in wild-type olfactory epithelium, centrioles were found clustered at each pole (**Fig 2B**). Likewise, clusters of centrioles were observed at opposite ends of the cell during anaphase, when phospho-H3 signal is more diffuse (**Fig 2C**). We also found mitotic cells in cryosections of developing olfactory epithelia from embryonic (E12.5) mice expressing eGFP-centrin2. Similar to what we observed in adult tissue, clusters of centrioles were found on either side of metaphase chromatin and at each pole of the cell during cleavage furrow formation (**Fig 2D and 2E–2E"**). These results demonstrate that OSN precursors are able to divide after centriole amplification in both adult and embryonic olfactory epithelium, and that both sister cells from a division can receive an amplified set of centrioles.

To determine when centriole amplification occurs within the OSN lineage, we conducted a secondary analysis of an existing single-cell RNA sequencing (scRNAseq) data set. Fletcher and colleagues sequenced cells from dissociated mouse olfactory epithelium and grouped cells into distinct cell states by Slingshot analysis [21,22]. Analysis of a suite of genes associated with cell cycle progression showed that progenitors known as globose basal cells (GBCs) and early immediate neuronal precursors (INPs) are likely mitotically active [21]. We specifically examined individual genes known to be up-regulated in association with DNA synthesis, including *Rrm2*, which encodes ribonucleotide reductase 2 [23]. We found that the mRNA for many of these genes was abundant only in cells in the GBC and INP1 states, consistent with these being mitotically active (**Fig 3A**). Next, we analyzed the scRNAseq data for the expression pattern of centriole-associated genes. In particular, we analyzed expression of *Plk4*, a gene whose expression is necessary for centriole duplication and whose up-regulation is a signature of centriole amplification in multiciliated cells [4]. Remarkably, the mRNA for Plk4 was strongly elevated in neuronally-fated INP1s and INP2s compared with the multipotent GBCs (**Fig 3B**). Plk4 drives centriole formation in conjunction with a binding partner, SCL/Tal1-interrupting locus gene (Stil) [13,15]. We found that the mRNA for Stil was also elevated in INP1s and INP2s (**Fig 3B**, **S3B Fig**). The mRNA for centrosomal protein 152 (Cep152), another binding partner of Plk4, also followed this pattern (**S3A Fig**) [24]. Given that up-regulation of Plk4 or Stil mRNA drives centriole rosette formation in cell culture and coincides with centriole amplification in multiciliated cells, we hypothesized that elevated Plk4 and Stil might mark the timing of centriole amplification in early INPs.

To determine whether elevated Plk4 and Stil mRNA levels correlate with the timing of centriole amplification, we used neuronal differentiation 1 (NeuroD1) as a marker of

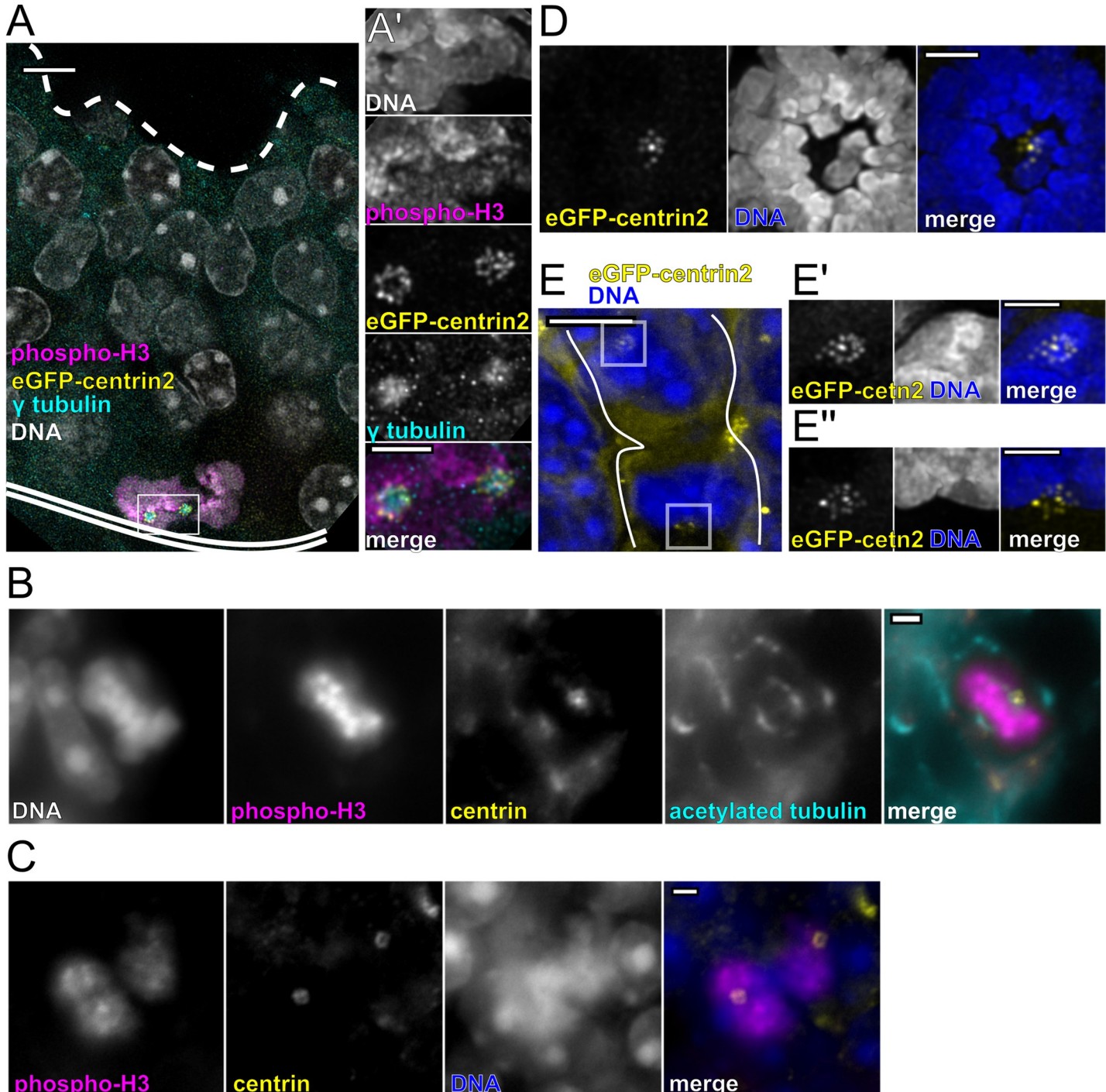

**Fig 2. Division of cells with amplified centrioles.** (A) Maximum projection image of immunofluorescence in olfactory epithelium cryosections from an adult mouse expressing eGFP-centrin2. Dashed line marks the apical surface of the olfactory epithelium. Double solid line marks the basal lamina. Box marks the location of the inset shown in A'. Scale bar = 5 μm. Inset (A') shows a deconvolved maximum projection of centrioles shown in panel A. eGFP-centrin2 and γ tubulin mark 2 clusters of centrioles. Phospho-H3 marks a cell in an early phase of mitosis. DAPI shows DNA condensed near the basal lamina and is excluded from the merge. Inset scale bar = 2 μm. (B) Immunofluorescence in cryosections of olfactory epithelium from a wild-type adult mouse. DAPI shows DNA condensed and aligned in metaphase with strong phospho-H3 colocalization. Centrin marks the distal ends of centrioles. In this single optical section, centriole clusters are shown at the spindle poles, marked by acetylated tubulin. DNA is excluded from the merge. Scale bar = 2 μm. (C) Single optical section image of immunofluorescence in olfactory epithelium from a wild-type adult mouse. Diffuse phospho-H3 marks a cell in anaphase. White arrows denote clusters of centrioles on opposite sides of the dividing cell. Scale bar = 2 μm. (D) Maximum projection of a deconvolved fluorescence image in olfactory epithelium from a mouse at embryonic stage E12.5. DAPI shows DNA condensed and aligned in metaphase. One spindle pole is shown with a cluster of centrioles marked by eGFP-centrin2. Scale bar = 2 μm. (E) Fluorescence image of a mouse at embryonic stage

E12.5. DAPI shows DNA decondensing to form 2 daughter cell nuclei. White lines show cell boundaries, including a cleavage furrow, as approximated by cytoplasmic eGFP-centrin2 resulting from overexpression. Boxes mark the location of inset images. Scale bar = 5 μm. The insets (E' and E") show deconvolved maximum projection images of centriole clusters in the top and bottom boxes, respectively. Centrioles are marked by eGFP-centrin2. Inset scale bars = 2 μm. See S2 Fig for additional details. Arl13b-mCherry, ADP-ribosylation-factor-like GTPase 13b; eGFP, enhanced green fluorescent protein; phospho-H3, phosphorylated histone 3.

developmental timing within the differentiation pathway for OSNs (S3C Fig). NeuroD1 is a transcription factor specifically up-regulated in early INPs, which are fated to become OSNs (Fig 3C) [21,25]. We identified OSN progenitors in sections of adult olfactory epithelium by their localization near the basal lamina and presence of nuclear NeuroD1 immunofluorescence signal. We found examples of cells with two centrioles and cells with more than two centrioles within the NeuroD1-positive progenitor population (Fig 3D and 3E). To avoid counting artifacts associated with sectioning and to improve imaging resolution, we used cells dissociated from olfactory epithelia of adult mice expressing eGFP-centrin2 to quantify centriole number in NeuroD1-positive precursors (Fig 3F and S3D Fig). We compared these counts to the number of centrioles per OSN imaged in septa from adult mice expressing eGFP-centrin2. As in cryosections, we found two groups within the NeuroD1-positive cells: A minority ($n = 4$) of cells that had only one or two visible centrioles, suggesting that they had not yet amplified centriole number, and a majority ($n = 36$) that had many more centrioles per cell (6 to 39 centrioles per cell). This distribution of centriole numbers suggests that centrioles are amplified during *NeuroD1* expression (S3C Fig). Together, our data support a model in which centriole amplification occurs during a transient state characterized by elevated Plk4 and Stil in early INP cells, which then go on to divide at least once to become OSNs (Fig 4).

## Discussion

OSNs in mammals have a configuration of centrioles and cilia that distinguishes them from most other cells. Building on a previous observation of centriole rosettes in the olfactory epithelium of embryonic mice [16], we have found that centrioles in both adult and embryonic olfactory epithelium can be amplified from the centrosome of the progenitor cell via centriole rosettes prior to cell division. This amplification occurs in neuronally fated progenitors and is correlated with increased expression of the centriole duplication proteins Plk4 and Stil.

One of the questions raised by these findings is how the final number of centrioles in mature OSNs is achieved. The simplest possibility is a that a single round of formation of new centrioles in a rosette around the two mother centrioles in a progenitor cell is sufficient. Mature OSNs had a mean centriole number of 15.7, with as many as 37 observed in a single cell, whereas the rosettes that we observed in the olfactory epithelium and in cultured cells overexpressing Plk4 had no more than eight centrioles, suggesting that this simple model cannot account for the total number. It is possible to form rosettes with more centrioles, similar to what has been observed in multiciliated cells lacking deuterosomes [10]. We did not observe such cases in olfactory epithelium, but we cannot rule out that it occurs. However, our observation of free centrioles in cells with rosettes suggests as alternatives that centrioles might form by de novo synthesis coincident with rosette formation or that new centrioles disengage from rosettes, allowing new centrioles to form continuously on the mother centriole. It is also possible that centriole amplification might occur in more than one cell cycle or after the final cell division in OSN differentiation. There is precedence for the latter in multiciliated epithelial cells, the only other widely studied example of centriole amplification in vertebrates, in which cells only amplify centrioles after division ceases [26].

Our results show that centriole amplification can occur in mitotically active progenitors of the olfactory epithelium. This is interesting because division with amplified centrosomes,

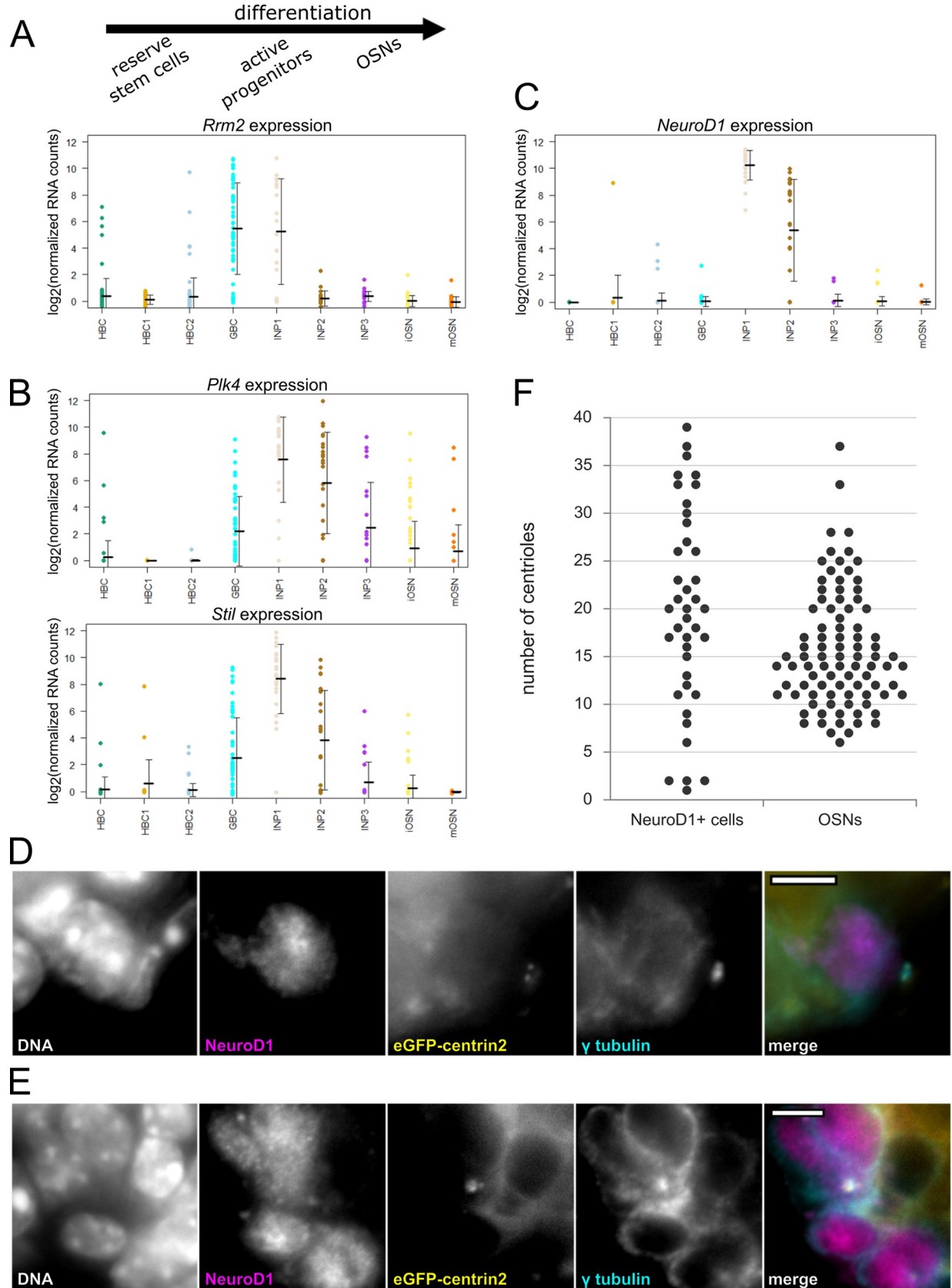

**Fig 3. *Plk4* and *Stil* RNA levels and centriole number in early immediate neuronal precursors in the olfactory epithelium.** (A-C) Secondary analysis of an existing single-cell RNA sequencing data set from Fletcher and colleagues compares RNA levels for specific genes across cell types in the olfactory epithelium [21]. The vertical axis shows average log$_2$(normalized RNA counts). Each dot represents one cell. The horizontal axis shows cell groups in the pseudotime lineage order determined by Fletcher and colleagues and is summarized at the top of panel A. (A) RNA levels for *Rrm2*, a gene specific to DNA synthesis in S phase. (B) RNA levels for genes that drive centriole formation *Plk4* and *Stil*. (C) RNA levels for *NeuroD1*, a transcription factor marking early immediate neuronal precursor cells. Center lines = mean. Error bars = standard deviation. See S2 Data for values. (D-E) Fluorescence images of olfactory epithelium from adult mice expressing eGFP-centrin2 and Arl13b-mCherry. Cryosections were stained with antibodies against NeuroD1 and γ tubulin and with DAPI to mark DNA. (D) A NeuroD1-positive cell with two centrioles. (E) A NeuroD1-positive cell with greater than two centrioles. DNA is excluded from the merge. Scale bars = 5 μm. (F) Comparison of centriole counts in different cell types from olfactory epithelium of adult mice expressing eGFP-centrin2 and Arl13b-mCherry. Centrioles in NeuroD1-positive cells were counted in dissociated olfactory epithelia ($N = 2$ mice, $n = 40$ cells). Centrioles in OSNs were counted by en face imaging of the apical surface of septum olfactory epithelia ($N = 6$ mice, $n = 90$ cells). Note the small population of NeuroD1-positive cells with unamplified centrioles. See S3 Data for centriole counts. See S3 Fig for additional details. Arl13b-mCherry, ADP-ribosylation-factor-like GTPase 13b; eGFP, enhanced green fluorescent protein; *NeuroD1*, neuronal differentiation 1; OSN, olfactory sensory neuron; *Plk4*, polo-like kinase 4; *Rrm2*, ribonucleotide reductase molecule 2; *Stil*, SCL/Tal1 interrupting locus gene.

mature centrioles that are able to nucleate microtubules, is considered to be detrimental because of the increased frequency of chromosome missegregation, particularly in the context of cancer [27,28]. Several features of the process in the OSN lineage mitigate the potential problem of mitosis with amplified centrioles. First, newly formed centrioles would not have matured by undergoing centriole-to-centrosome conversion, which promotes their ability to nucleate microtubules and form spindle poles in the mitosis [29]. Even if amplified centrioles had matured, for example, by amplification occurring over more than one cell cycle, known mechanisms could enforce bipolar spindle formation, for example, by kinesin family member KifC1/HSET-dependent centriole clustering [28]. Second, many of the amplified centrioles are contained within rosettes with only a single mother centriole (**S1D1–S1D4 Fig**) that would be competent to nucleate microtubules. Indeed, we found that each rosette in this case had only a single focus of γ-tubulin (**Fig 2A', S2A' Fig**). This is similar to what appears to occur in spermatogenesis in some snails, during which cells with rosettes go through meiosis [30,31]. Cosenza and colleagues showed that the fidelity of mitosis in cells with overexpression-induced rosettes is sensitive to asymmetry in the number of daughter centrioles per rosette [32], although it is unknown whether this phenomenon plays a role in the OSN lineage.

We showed that the transcripts for key proteins in centriole duplication, Plk4 and Stil, are transiently up-regulated during OSN differentiation. OSN differentiation closely resembles that of multiciliated epithelial cells, except that no deuterosomes are found in cells amplifying centrioles (**Fig 1D**). In multiciliated epithelial cells, up-regulation of *Plk4* and other centriole-associated genes is a signature of centriole amplification via deuterosomes and rosettes during differentiation [4]. Up-regulation is not only relevant to the timing of centriole amplification though. Plk4 and Stil can each drive rosette formation upon overexpression in tissue culture [11,15]. This raises the question of whether this up-regulation is sufficient to coordinate the formation of centriole rosettes in the olfactory epithelium. Besides *Plk4* and *Stil*, the only other centriole-associated gene that followed the same pattern of sharply increased RNA levels in early INP cells was *Cep152*. Some other genes necessary for centriole duplication [12,33] show modest changes in expression between cell types (see **S3A Fig**) but none as striking as *Plk4*, *Stil*, and *Cep152*. Interestingly, Cep152 protein is known to be necessary for anchoring Plk4 and Stil at the mother centriole [24], but increased Cep152 has not been associated with rosette formation. Additionally, how *Plk4* and *Stil* transcription or RNA half-life is increased in early INPs remains unclear.

In summary, our work characterizes a differentiation program in which centriole amplification and cell division occur as a normal part of development and organ maintenance. These findings highlight the robustness of mitotic division to alterations in centriole number, adding

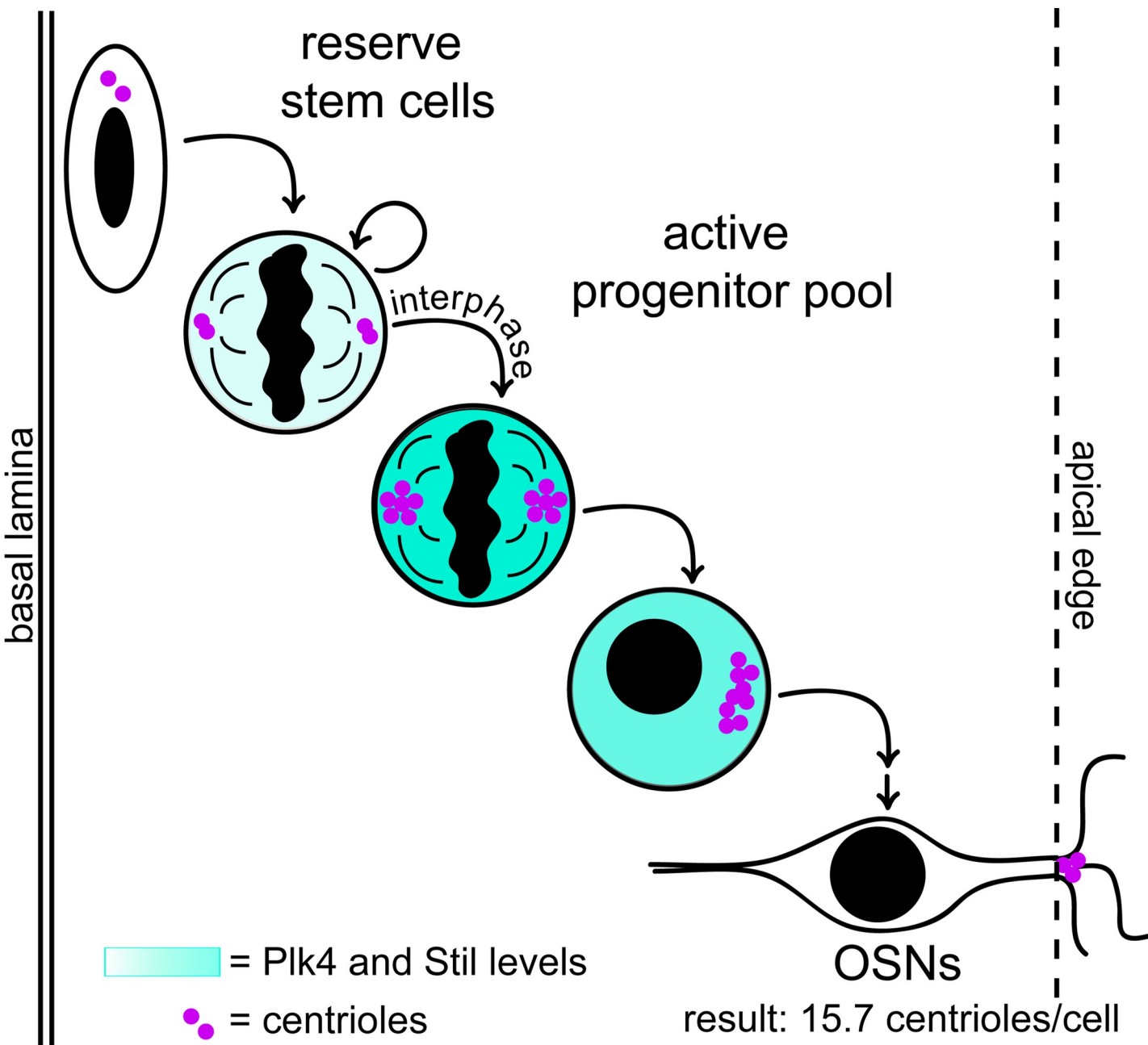

**Fig 4. Summary of centriole amplification in the olfactory epithelium.** OSN, olfactory sensory neuron; Plk4, polo-like kinase 4.

to existing evidence from disease states, such as cancer, and the flexibility of centriole amplification programs to meet the needs of specific cell types.

## Methods

### Ethics statement

This study uses samples from mice. All animal procedures in this study were approved by the Stanford University Administrative Panel for Laboratory Animal Care (SUAPLAC protocol 11659) and carried out according to SUAPLAC guidelines.

**Table 1. Mice and cells used.**

| Experiment | Samples used |
|---|---|
| TEM | 3-week-old male wild-type CD1 mice from Charles River |
| quantification of centrioles per OSN and centrioles per NeuroD1-positive progenitor | adult (1 to 14 months) TgCAG-Arl13b-mCherry, eGFP-centrin2 mice from Jackson Laboratories [17] |
| immunofluorescent staining in olfactory epithelium | adult (7 to 12 months) C57BL/6 mice and TgCAG-Arl13b-mCherry, eGFP-centrin2 mice from Jackson Laboratories [17] embryonic stage E12.5 TgCAG-Arl13b-mCherry, eGFP-centrin2 mice from Jackson Laboratories [17] |
| centriole area analysis in cell culture | hTert RPE-1 TetOn-Plk4-Flag, eGFP-centrin2 cells (a gift from Bryan Tsou) |

Arl13b-mCherry, ADP-ribosylation-factor-like GTPase 13b; eGFP, enhanced green fluorescent protein; NeuroD1, neuronal differentiation 1; OSN, olfactory sensory neuron; RPE-1, retinal pigmented epithelial; TEM, transmission electron microscopy.

## Antibodies used

Primary antibodies used for immunofluorescent staining are listed in Table 2. AlexaFluor-conjugated secondary antibodies (Thermo-Fisher) were diluted 1:1,000.

## Transmission electron microscopy

Mice (Table 1) were euthanized by $CO_2$ in accordance with Stanford's APLAC guidelines. Facial bones were removed in a dish of cold Tyrode's solution (140 mM NaCl, 5 mM KCl, 10 mM HEPES, 1 mM $CaCl_2$, 1 mM $MgCl_2$, 1 mM sodium pyruvate, 10 mM glucose in $ddH_2O$), as in other reports [34], and turbinate scrolls were mechanically separated from septa. Epithelia from turbinate scrolls were removed mechanically and fixed immediately in a solution of 2% glutaraldehyde and 4% PFA in 0.1M Na cacodylate buffer for 3 to 4 hours at 4˚C. Samples were then rotated in a 1% solution of $OsO_4$ for 1 hour at room temperature, washed 4 times gently in water, then rotated in a 1% solution of uranyl acetate overnight at 4˚C. Samples were then dehydrated in a graded ethanol series (30%, 50%, 70%, 95%, 100%, 100%) for 15 to 20 minutes per step, rotating at room temperature. Samples were washed twice for 10 minutes each in propylene oxide (PO), then embedded through a graded PO:EMBED resin series (2:1 for 1 hour, 1:1 for 1 hour, 1:2 overnight). Samples were then rotated in pure EPON with lids

**Table 2. Primary antibodies used.**

| Target | Source | Dilution | Treatment |
|---|---|---|---|
| β tubulin III | BioLegend, clone TuJ1 | 1:2,000 | none needed |
| γ tubulin | Sigma-Adrich, clone GTU88 | 1:1,000 | none needed |
| centrin (all isotypes) | EMD Millipore, clone 20H5 | 1:5,000 | 10-minute pretreatment in 0.5% (w/v) sodium dodecyl sulfate in PBS for olfactory epithelium |
| phospho-H3, Ser10 | Cell Signaling, #53348T | 1:200 | none needed |
| GFP | Invitrogen, #A-11120 | 1:1,000 | none needed |
| Sass6 | Santa Cruz Biotech, #91.390.21 | 1 µg/mL | requires cell fixation in −20˚C methanol |
| NeuroD1 | Proteintech, #12081-1-AP | 1:100 | 10-minute pretreatment in 0.5% (w/v) sodium dodecyl sulfate in PBS, requires overnight incubation in primary antibody |

NeuroD1, neuronal differentiation 1, w/v, weight in grams per 100 mL volume.

open for 5 hours to evaporate remaining PO before embedding in molds at 50˚C for 4 days. Semithin sections were taken and imaged on a dissecting scope to find samples in the correct orientation. Sections of 80 nm thickness were treated with uranyl acetate and mounted on grids before imaging on a JEOL JEM-1400. Samples were prepared from two separate animals, and rosettes and free centrioles were observed in both. Images were processed in Fiji [35], and images in panels that include low-magnification images were rotated such that the basal lamina is at the bottom.

## Immunofluorescence staining of cryosections

Olfactory epithelia were dissected as described for TEM. Whole olfactory epithelia, turbinate epithelia, or E12.5 embryo heads from mice (Table 1) were fixed immediately in 4% PFA in PBS at 4˚C for 3 to 24 hours. Samples were then washed in PBS and stored at 4˚C. Before mounting, samples were equilibrated in 1 to 5 mL of 30% sucrose solution in water for a minimum of 12 hours at 4˚C. Samples were embedded in OCT compound (Sakura Tissue-Tek) on dry ice and stored at −80˚C. Embedded samples were sectioned at 8 to 14 µm on a Leica cryostat and adhered to charged slides by drying at room temperature for approximately 1 hour. Slides were stored with drying pearls (Thermo-Fisher) at −80˚C and thawed under desiccation no more than twice. Samples were pretreated as needed (see antibody summary chart), then rehydrated and blocked for 0.5 to 4 hours in 5% milk in 0.1% Triton-x 100 that had been spun in a tabletop centrifuge to pellet undissolved milk particles. Slides were incubated in primary antibody for approximately 3 hours, washed in PBS, incubated in secondary antibody for approximately 1 hour, washed in PBS, incubated in DAPI for 1 to 5 minutes, washed in PBS, and mounted in MOWIOL. The adult sample shown in Fig 2A and all embryonic samples were imaged on a Leica SP8 spinning disk confocal microscope using the Leica Application Suite X (LAS X) software, and insets were deconvolved using HyVolution. All other adult samples were imaged on a Zeiss inverted widefield microscope using MicroManager [36]. Images were processed in Fiji [35]. Images from the widefield microscope were deconvolved using the Iterative Deconvolve plug-in [37] and theoretically generated point spread functions (Diffraction PSF 3D). Images were processed in Fiji [35], and images in panels that include low-magnification images were rotated such that the basal lamina is at the bottom. For images with high background, contrast in the representative images was adjusted uniformly across the image such that the area outside of cells was black and areas of high signal were just below saturation. Each immunofluorescent staining procedure was performed at least three times ($N \geq 3$) with samples taken from at least two separate animals.

## Analysis of centriole structure area

To quantify the area of centriole structures, samples were imaged on a Zeiss inverted widefield microscope. For proof of concept, centrioles from RPE-1 cells processed for immunofluorescence staining (see procedure described next) were imaged in z-stacks with 0.5-µm steps to include all centrioles in the field of view ($N = 3$ replicates from cell seeding through staining). A total of 30 images of each condition were taken and processed in Fiji [35]. Z-stacks were converted into a maximum projection image, and the green channel was deconvolved using the Iterative Deconvolve plug-in [37]. Engaged structures were selected based on the presence of anti-Sass6 immunofluorescence signal between adjacent GFP puncta (visualized by immunofluorescence for GFP, and rosettes were defined as structures with at least three GFP puncta. We measured the area of GFP fluorescence in structures meeting these criteria and normalized all measurements such that the average area of centriole pairs was exactly 2. The normalized fluorescence area had approximately a 1:1 ratio with actual centriole number (0.9208),

demonstrating that it is an appropriate approximation for centriole number (**S2C Fig**). Next, we used this method to assess centriole structures from the olfactory epithelium (**Fig 3C**). We estimated the probability density distribution of centriole pair areas to be a Gaussian function. We used this distribution to estimate a cutoff area (0.7085 μm²) above which a structure has less than 1% probability of belonging to the centriole pairs data set. As a proof of concept, 73.0% of rosettes measured in cell culture were above this cutoff. For olfactory epithelia, single-plane images were used for analysis shown here, though similar results were obtained with z-stacks. We applied the cutoff to mitotic cells of the olfactory epithelium because, in contrast to S-phase cells, mitotic cells' centrioles separate in preparation for spindle formation, reducing overlap and making structures more amenable to measurement. Images of anti-centrin immunofluorescence signal in mitotic cells in cryosections from adult mice were deconvolved using the Iterative Deconvolve plug-in, and area was measured by outlining puncta in the anti-centrin channel. Images in which centriole pairs were clearly visible were categorized as such. All other images were categorized as "nonpair" structures. Immunofluorescent staining was performed five times, and all mitotic cells centrioles that could be imaged were included. A total of 87.2% of area measurements in this group fell above the cutoff. Probability calculations were performed in R (code available at https://github.com/katieching/CentrioleAreaAnalaysis). Dot plots, means, and standard deviation values were generated with Statistika [38]. Linear regression was carried out using Excel. Data are available in S1 Data.

## Overexpression of Plk4 in cell culture

RPE-1 cells (Table 1) were cultured in DMEM/F-12 (Corning #MT-10-092-CV) with 10% Cosmic Calf Serum (GE Healthcare #SH30087.04) and periodically tested by PCR for mycoplasma contamination. Stock cultures were selected by hygromycin B (Thermo-Fisher #10687010) prior to the start of experiments. Cells were seeded to be 70%–80% confluent at the start of the experiment. S-phase arrest was initiated by incubating cells in 2 mM thymidine. After 24 hours, media was replaced with new media containing 1 μg/mL doxycycline and 2 mM thymidine to induce overexpression or with 2 mM thymidine and DMSO for the control condition. After 24 additional hours, cells were washed in PBS, fixed for 20 minutes in methanol at −20˚C, washed again in PBS, and stored at 4˚C.

## Immunofluorescence staining of RPE-1 cells

Cells cultured on poly-L-lysine-coated coverslips were fixed in methanol at −20˚C for 20 minutes and washed in PBS. Samples were blocked for a minimum of 30 minutes in 5% dry milk in 0.1% Triton-x 100 that had been spun to pellet undissolved milk particles. Samples were then washed 3 times in PBS, incubated with primary antibodies for 1 to 2 hours at room temperature, washed 3 times in PBS, incubated with secondary antibodies for 0.5 to 2 hours, washed 3 times in PBS, incubated with DAPI for 5 minutes, washed 3 times in PBS, and mounted in MOWIOL. Cells were imaged on a Zeiss inverted widefield microscope with MicroManager [36]. Images were processed in Fiji [35].

## Secondary analysis of scRNAseq data

Single-cell RNAseq data from Fletcher and colleagues in 2017 [21] were obtained as a .rda file from the authors and are also available using the authors' accession number GEO, GSE95601. Cells were pooled for average RNA levels based on categories determined by Fletcher and colleagues. Analyses for single gene expression and coexpression were performed in R (code available at https://github.com/katieching/RNAseq). Mean and standard deviation values are available in S2 Data.

## Quantification of centrioles in OSNs

Samples were dissected as described for TEM, except that dissections were performed in PBS. Septa were immediately transferred to 4% paraformaldehyde (PFA) in PBS and fixed for 3 to 24 hours at 4˚C. Septa were washed and stored in PBS at 4˚C. For imaging, septa were mounted in a chamber of double-sided tape on glass slides with SlowFade Gold mountant (Invitrogen) and high-precision 1.5 weight coverslips (Deckglässer) sealed with nail polish. Samples were imaged on a Leica SP8 scanning confocal microscope. For each sample, 5 fields of view were spaced approximately evenly along the anterior–posterior axis of the olfactory epithelium. Within each field of view, the cell at the center of each quadrant of the field was imaged such that the z-stack included all centrioles within the dendritic knob. The lowest-quality image from each field of view was excluded from the analysis, giving 15 cells per animal ($N$ = 6 animals, $n$ = 90 cells total). Individual dendrites were identified by their tightly clustered centrioles. Image stacks were processed by semiautomated detection in the program Imaris x64 9.2.1 (Oxford Instruments) using the Surfaces function and separating touching objects by seed points of 0.3-μm diameter. Dot plots were generated using Statistika [38]. Data are available in S3 Data.

## Olfactory epithelium dissociation

The turbinate region of olfactory epithelia was dissected in cold Tyrode's solution, as described for TEM. Samples were incubated in 1 to 2 mL of 0.25% trypsin (Thermo-Fisher, #MT-25-053-CI) and minced with a feather scalpel periodically, between incubations at 37˚C, for 10 to 15 minutes in total. Trypsin was inactivated by adding 10 mL of DMEM with 10% serum. Samples were poured over a 40-μm cell strainer to remove bone fragments and other large debris. Samples were spun at 800$g$ for 5 minutes to pellet, washed in PBS, and spun again. Samples were resuspended and fixed in 4% PFA in PBS overnight at 4˚C, then washed and stored in PBS at 4˚C.

## Quantification of centriole number in progenitor cells

Dissociated samples ($N$ = 2) were stained to identify NeuroD1-positive progenitors by first spinning at 8,000 rpm for 2 minutes in a tabletop centrifuge to remove PBS, then pretreating by resuspending in 0.5% (w/v) SDS in water for 1 minute. Cells were spun to remove SDS, then resuspended and blocked for 0.5 to 1 hour at 4˚C in a solution of 5% dry milk in 0.1% Triton-x 100 that had been spun to remove undissolved milk particles. Samples were washed in PBS, incubated in primary antibody overnight at 4˚C, washed in PBS, incubated in secondary antibody for 30 minutes at room temperature, washed in PBS, incubated in DAPI solution for 2 minutes, washed in PBS, then resuspended in MOWIOL. Samples were mounted with 1.5-weight coverslips, and centrioles were imaged ($n$ = 20 cells per sample, $n$ = 40 total) and counted by the same method as quantification of centrioles in OSNs. Data are available in S3 Data.

## Supporting information

**S1 Fig. Coexistence of engaged and nonengaged centrioles.** (A) TEM image of wild-type adult mouse olfactory epithelium. Dashed line marks the apical surface of the olfactory epithelium. Double solid line marks the basal lamina. Box marks the location and orientation of the inset shown in the panel to the right. Scale bar = 10 μm. The inset shows a centrosome and primary cilium pseudocolored purple and cyan, respectively. Inset scale bar = 0.5 μm. (B) Inset from a maximum projection fluorescence image of embryonic olfactory epithelium at E12.5 in

mice expressing eGFP-centrin2 to mark centrioles, shown in Fig 1C. Deconvolved images show two rosette-like centriole clusters and separate puncta positive for centriole markers eGFP-centrin2 and γ tubulin. Scale bar = 2 μm. (C) Inset from a single optical section of embryonic olfactory epithelium at E12.5 in mice expressing eGFP-centrin2 to mark centrioles, shown in Fig 1C and S1B. Arrows mark the location of a rosette (shown in panel B) at the base of a primary cilium in a cell that is positive for β tubulin III. Asterisks mark a nearby cell that is negative for β tubulin III. Labels denote method of detection. Scale bar = 5 μm. (D1-D4) TEM images from serial sections of olfactory epithelium from a wild-type adult mouse. R1, R2 denote centriole rosettes, identified by morphology. C1-5 denote centrioles not associated with rosettes. Note that both mother centrioles in panel D1 have accessory structures, confirming that both rosettes form on centrioles that existed for at least one previous cell cycle. Scale bar = 1 μm. TEM, transmission electron microscopy.
(TIF)

**S2 Fig. Division of cells with amplified centrioles in the olfactory epithelium.** (A) Immunofluorescence in cryosections of olfactory epithelium from a wild-type adult mouse. Punctate nuclear PCNA marks a cell in S phase, whereas nearby nuclei are PCNA-negative. Dashed line marks the apical surface of the olfactory epithelium. Double solid line marks the basal lamina. Box marks the location of the inset. Scale bar = 20 μm. In the inset (A'), DAPI marks DNA of the S-phase cell, identified by punctate PCNA. CP110 marks the distal ends of centrioles and γ tubulin marks centrosomes. In this single optical section, daughter centrioles are visible as rings around γ tubulin foci, consistent with rosette formation. For clarity, the DNA panel is excluded from the merge. Inset scale bar = 2 μm. (B) Analysis of eGFP-centrin2 fluorescence area in mitotic cells in the olfactory epithelium. The pair (culture) column ($N = 3$, $n = 208$) shows measurements of centriole pairs in RPE-1 cells, which were used to set a threshold of 0.7085 μm$^2$ (purple line), above which area measurements have <1% probability of belonging to the centriole pairs data set. The rosette (culture) column ($N = 3$, $n = 115$) shows measurements of centriole rosettes in cells overexpressing Plk4, 73.0% of which are above the threshold. The mitosis pair (OE) column ($N = 5$, $n = 29$) shows measurements of centriole pairs in adult olfactory epithelium, all of which fall below the threshold. The mitosis nonpair (OE) column ($N = 5$, $n = 46$) shows measurements of centriole structures which could not be definitively classified as pairs. A total of 87.2% are above the threshold. See S1 Data for measurement values. (C) Plot of anti-GFP fluorescence area against centriole number in cell culture. Immunofluorescence images were taken of hTert RPE-1 TetON-Plk4, eGFP-centrin2 cells with and without doxycycline induction. Anti-GFP fluorescence area of Sass6-positive structures was measured, and puncta were counted by eye. A line of best fit was generated in Microsoft Excel. The slope of the line is 0.9208, showing an approximately linear relationship between centrin fluorescence area and centriole number. See S1 Data for measurement values. eGFP, enhanced green fluorescent protein; OE, olfactory epithelium; PCNA, proliferating cell nuclear antigen; Plk4, polo-like kinase 4.
(TIF)

**S3 Fig. RNA levels in scRNAseq data and images of a NeuroD1-positive cell.** (A) Secondary analysis of an existing single-cell RNA sequencing data set from Fletcher and colleagues (2017) compares RNA levels for specific genes across cell types in the olfactory epithelium. The vertical axis shows log$_2$(normalized RNA counts). Cell groups are ordered by pseudotime along the horizontal axis. Plots show RNA levels for *Cep152*, *Cep192*, and *Sass6*, genes required for centriole duplication (see Hatch and colleagues, 2010; Gomez-Ferreria and colleagues, 2007; Leidel and colleagues, 2005). Dots represent individual cells. Center lines = mean, and error bars = standard deviation. See S2 Data for values. (B-C) Secondary analysis of data from

Fletcher and colleagues, 2017 shows coexpression of genes. Each dot represents a single cell. Both axes show $\log_2$(normalized RNA counts) (B) Plot shows RNA levels for *Plk4* and *Stil*, centriole-associated genes known to drive rosette formation in cell culture. Points in the upper right corner are cells that express both genes at high levels. These are INP1 and INP2 cells (see A for color coding). (C) Plot shows RNA levels for *NeuroD1*, a transcription factor that marks INP1 and INP2 cells, and *Plk4*. Points in the upper right corner are cells that express both genes. These show that individual INP1/2 cells express high levels of *Plk4*. (D) A fluorescence image of NeuroD1-positive cells in dissociated olfactory epithelium. Note that this image includes other nuclei that are NeuroD1-negative. Boxes in the eGFP-centrin2 panel mark the locations of insets. DNA is excluded from the merge. Scale bar = 5 μm. The insets (1, 2) show deconvolved maximum projection images of centrioles, marked by eGFP-centrin2. Inset scale bars = 2 μm. Cep152, centrosomal protein 152; Cep192, centrosomal protein 192; eGFP, enhanced green fluorescent protein; INP, immediate neuronal precursor; NeuroD1, neuronal differentiation 1; Plk4, polo-like kinase 4; Sass6, spindle assembly abnormal protein 6; scRNA-seq, single-cell RNA sequencing; Stil, SCL/Tal interrupting locus gene.
(TIF)

**S1 Data. Measurements of centriole area by anti-centrin fluorescence.**
(CSV)

**S2 Data. Mean and Standard Deviation Values for Secondary Analysis of scRNAseq Data from Fletcher and colleagues, 2017.** Corresponding plots are shown in Fig 3A–3C and S3A Fig. scRNAseq, single-cell RNA sequencing.
(XLSX)

**S3 Data. Number of centrioles in NeuroD1-positive progenitor cells and OSNs.** NeuroD1, neuronal differentiation 1; OSN, olfactory sensory neuron.
(CSV)

## Acknowledgments

We thank John Ngai, Russell Fletcher, and Diya Das for helpful suggestions and feedback on the secondary analysis of their scRNAseq data, Eszter Vladar for helpful discussions, Emily Kolenbrander Ho for feedback on the manuscript, Mary Mirvis for assistance with image analysis, and John Perrino and the Stanford Cell Sciences Imaging Facility for help with TEM. We thank Stanford Bio-X for access to microscopy resources and members of the Stearns lab for helpful feedback and suggestions.

## Author Contributions

**Conceptualization:** Kaitlin Ching, Tim Stearns.

**Data curation:** Kaitlin Ching.

**Formal analysis:** Kaitlin Ching.

**Funding acquisition:** Tim Stearns.

**Investigation:** Kaitlin Ching.

**Methodology:** Kaitlin Ching.

**Project administration:** Tim Stearns.

**Supervision:** Tim Stearns.

**Validation:** Kaitlin Ching, Tim Stearns.

**Visualization:** Kaitlin Ching.

**Writing – original draft:** Kaitlin Ching.

**Writing – review & editing:** Kaitlin Ching, Tim Stearns.

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
