## [Editor Report · Decision Letter 0]

9 Jun 2020

Dear Dr Stearns, 

Thank you for submitting your manuscript entitled "Centrioles are amplified in cycling progenitors of olfactory sensory neurons" for consideration as a Short Reports by PLOS Biology.

Your manuscript has now been evaluated by the PLOS Biology editorial staff as well as by an academic editor with relevant expertise and I am writing to let you know that we would like to send your submission out for external peer review.

Please re-submit your manuscript within two working days, i.e. by Jun 11 2020 11:59PM.

Kind regards,

Di Jiang

PLOS Biology

---

## [Decision Letter · Decision Letter 1]

14 Jul 2020

Dear Dr Stearns,

Thank you very much for submitting your manuscript "Centrioles are amplified in cycling progenitors of olfactory sensory neurons" for consideration as a Short Reports by PLOS Biology. Your paper was evaluated by the PLOS Biology editors as well as by an Academic Editor with relevant expertise and by three independent reviewers. 

Based on the reviews, we will probably accept this manuscript for publication, assuming that you will modify the manuscript to address the points raised by the reviewers including the concerns about image quality and the Discussion. Please also make sure to address the data and other policy-related requests noted at the end of this email.

We expect to receive your revised manuscript within two weeks. Your revisions should address the specific points made by each reviewer. Please provide a 'Response to Reviewers' file which presents a point-by-point response to all of the reviewers' comments, and indicates the changes made to the manuscript.

In addition to the revisions and before we will be able to formally accept your manuscript and consider it "in press", we also need to ensure that your article conforms to our guidelines. A member of our team will be in touch shortly with a set of requests. As we can't proceed until these requirements are met, your swift response will help prevent delays to publication.

*Copyediting*

*Published Peer Review History*

*Early Version*

*Submitting Your Revision*

Sincerely,

Di Jiang, PhD 

Senior Editor

PLOS Biology

ETHICS STATEMENT:

-- Please create a separate subsection entitled "Ethics Statement" and place it in the beginning of the Methods section. Please include all relevant information described below including ethe approval number. 

-- Please include the full name of the IACUC/ethics committee that reviewed and approved the animal care and use protocol/permit/project license. Please also include an approval number.

-- Please include the specific national or international regulations/guidelines to which your animal care and use protocol adhered. Please note that institutional or accreditation organization guidelines (such as AAALAC) do not meet this requirement.

-- Please include information about the form of consent (written/oral) given for research involving human participants. All research involving human participants must have been approved by the authors' Institutional Review Board (IRB) or an equivalent committee, and all clinical investigation must have been conducted according to the principles expressed in the Declaration of Helsinki.

DATA POLICY:

Regardless of the method selected, please ensure that you provide the individual numerical values that underlie the summary data displayed in the following figure panels as they are essential for readers to assess your analysis and to reproduce it: Figures 3ABCF, S2BC, S3ABC. NOTE: the numerical data provided should include all replicates AND the way in which the plotted mean and errors were derived (it should not present only the mean/average values).

Reviewer remarks:

Reviewer #1: The vast majority of cycling cells in our bodies maintain centriole number at exactly two or four copies per cell to preserve cell division fidelity. However, multiciliated epithelial cells must produce hundreds of centrioles to nucleate many motile cilia and transport vital fluids. The current view is that massive centriole production in these terminally differentiated cells is enabled by the formation of cell-type-specific organelles called deuterosomes, which supplement the centrosome by providing additional sites for centriole nucleation. 

Interestingly, centriole number in olfactory neurons lies between the well-studied examples of cycling cells and multiciliated epithelial cells: each olfactory neuron creates 10-20 non-motile cilia that house the odorant receptors required for odorant sensing. To nucleate multiple cilia, olfactory neurons must amplify a similar number of centrioles, but how they achieve this feat has received little attention. 

In this manuscript, Ching et al., provide insight into the mechanisms of centriole amplification in olfactory neurons. The authors show that centrioles begin amplification in olfactory neuron progenitor cells where rosettes of newly created procentrioles form around the two-parent centrioles. A small number of free centrioles were also observed nearby these rosettes. The centriole rosettes remain clustered at the pole of the dividing progenitor cells so that each of the sister cells receives a collection of the amplified centrioles. This provides strong evidence that centriole amplification naturally occurs in the mitotically active progenitors in the olfactory epithelium. Analysis of single-cell RNA sequencing data revealed that cycling progenitors upregulated the mRNA levels of Plk4, STIL, and CEP152, which are critical upstream regulators of centriole biogenesis. CEP152 is a centriole receptor for the Plk4 kinase, while STIL is an activator of Plk4. This suggests that increased levels of Plk4 kinase activity are likely to help drive rosette formation in olfactory neuron progenitor cells.

This is a clearly presented manuscript that provides new insight into an important biological question that has been largely overlooked. I found the data to be compelling and the main conclusions to be well justified. The manuscript reports one of the few examples of a cell division occurring following a developmental pathway that leads to the amplification of centriole numbers - this presumably also occurs in liver cells, which often become polyploid with age. This study raises several important questions. For instance, although the generation of two rosettes increases centriole number, this alone is not sufficient to achieve the final number of centrioles required in olfactory neurons. The authors clearly discuss several possibilities for how to account this observation, but further detailed analysis would not be straight forward and is beyond the scope of the current study. Overall, I felt this was an excellent manuscript that will be of interest to the community of biologists studying centriole biogenesis and multiciliated cells. 

This is the second time I have reviewed this excellent paper and the authors already addressed my prior concerns. I have only a few minor comments that the authors may wish to consider. 

* I think it would be valuable to show some of the RNA seq data for the other key regulators of centriole biogenesis, such as SAS6 or CEP192. I understand these proteins do not change, but that might be good to show. It would also be interesting to know how many other genes show striking upregulation in these cell types.

* Is there any information on P53 levels/activity in the cells that have an amplified centriole content? Do the authors observe upregulation of P21 or other P53 target genes for example?

Reviewer #2: The mechanism governing the assembly of multiple basal bodies (centrioles) in olfactory sensory neurons (OSNs) has not previously been determined. Previous studies documented that terminally differentiated OSNs harbor approximately two dozen of basal bodies. Such basal bodies are formed in a rosette-like fashion, where multiple new centrioles associate with parental centrioles, or they seem to form in isolation from parent centrioles (de novo) PMID: 1141050. 

In this work, Ching and Stearns use light and electron microscopy to determine how and when multiple centrioles are generated within olfactory epithelium of mice. In addition, the authors explore the levels of transcripts of various centriole initiators and find that centrioles amplify in immediate neuronal precursor cells with elevated mRNA of Plk4 and STIL, two key centriole initiators. Interestingly, the authors also show that OSN precursors which harbor amplified centrioles undergo additional cell division clustering amplified centrioles on their poles. 

The paper explores an important topic and the findings could be interesting to the wide readership of PLOS biology. The drawback is that most imaging data is presented in a really poor quality. It is possible that the issue is only related to PDF presentation, but regardless of the reason, this issue impeded proper evaluation of the data. 

This work could be suitable for publication in PLOS biology after the issues listed below are addressed and the image quality is improved. 

Specific concerns: 

Point 1. Fig. 1C. beta tubulin III (a neuronal marker) and Arl13b (a ciliary marker) are both immunolabeled in the same color. Due to the poor image quality, it is difficult to understand which signal is which and why was this combination of markers used? Also, it is almost impossible to understand to which cell amplified centrioles belong to. 

Point 2. Fig. S2. Image quality is again a big issue. The content of the inserts in A' s is rotated with respect to the original low image panels in this figure (and throughout the manuscript on multiple places). This is not logical and could confuse readers. I would suggest that this is corrected. 

Point 3. Figure 2 A: DNA condensation near the basal lamina is not obvious. Provide clearer images. Please keep the same orientation of centrioles in A and in A'. 

Point 4. Figure 2 B. It is unclear why was Poly E tubulin used. Or is it gamma tubulin as stated in the legend? 

Point 5. Figure 2 D. Based on a poor overlap between DNA and phosphor-H3 staining, it is not clear that the image represents anaphase. DNA is smeared all over the place. The same comment can be addressed to Fig. 2 E. where the DNA pattern indicates that two nuclei are clearly out of mitosis and with decondensed DNA. Yet the cell boundaries drawn in white imply that these two nuclei are not yet separated (telophase?) and in the text (line 119they are describes to be "later in division" (Fig. 2C, E-E'). 

Point 6. Abstract, Line 29. The authors suggest that the cells with amplified centrioles divide one or more times to become OSNs. But there is no evidence that these cells divide more than once. Would that imply that cells should have more than 2 mature centrioles? This should be experimentally tested if the authors think that it is a possibility. 

Point 7. The authors might want to revisit discussion. It feels disproportionally long and at times unnecessarily speculative. For instance: 

Line 176 - 198: The authors estimate that the number of centrioles formed per mother centriole is 8 and question how cells reach a greater and "desired" number of centrioles. I am not sure that I understand the argument about the "desired" number of centrioles in OSNs since mature OSNs harbor a wide range of centrioles in mature OSNs (6-37, line 75). So, the number of centrioles in mature OSNs seems random. It would also agree with the variable number of C1-GFP foci detected in various cells (Fig. 2A', E', E'', S3). In addition, it is evident that all Centrin-GFP signals are not in rosette-like configurations and adjacent to mother centrioles. This should not be surprising given that even in cycling cells after Plk4 overexpression the number of centrioles is random and their orientation with respect to mother centrioles is not always preserved. The authors also speculate that centrioles could undergo a second round of amplification, but this should be tested rather than speculated. 

Line 200: The authors discuss possible detrimental consequences of centriole amplification on mitosis of progenitor cells. However, the presence of rosettes or de novo formed centrioles wont necessary lead to mitotic issues. Such issues usually occur if cells contain multiple mature centrioles (which are not detected in progenitor OSNs). The authors do not provide any data for or against possible mitotic issues in cells harboring amplified centrioles, yet, they discuss this is a great length and provide possible explanations why there might be no issues with mitosis. 

Point 8. Line 166-167: Centriolar rosettes and single centrioles in OSNs have been documented before and citation would be appropriate. 

Point 9. A scheme, Figure 4. It reads like there is a direct centriole amplification from one mitosis to the next, one interphase should be added in between.

Reviewer #3: The manuscript by Ching et al reports the in vivo characterization of how centriole amplification is conducted in the lineage of progenitors that eventually become the multiciliated, olfactory sensory neurons (OSNs). They found that centrioles are amplified in S phase in a type of cycling cells in the lineage called immediate neuronal precursors (INPs), using the amplification program known as the rosette formation. The INP cells with amplified numbers of centrioles will go on to mitosis and divide at least once before differentiating into OSNs where the inherited, numerous centrioles will be used for ciliogenesis. 

The study clearly figures out the process through which the numerous centrioles inherited by OSNs are produced and delivered during differentiation. It is interesting and important. I have no major concern with the data and conclusion. 

Minor issue:

There is only one point I would like to suggest to authors, which is to examine when/how these centrioles that are produced in cycling cells eventually acquire the appendages for ciliogenesis. Appendage formation is one of the final steps in centriole maturation process that in cycling cells normally takes ~1.5 cell cycle to finish (i.e. must go through two runs of mitosis), but in postmitotic cells, it can happen in the same cell cycle phase in which centrioles are born. It will be intriguing to know if centrioles inherited by OSNs also need to go through two or just one run of mitosis to be fully matured. Perhaps the authors can separately check NeuroD+ and NeuroD- cells carrying numerous centrioles for distal appendage markers? It could be interesting but not necessary for the paper as is.

---

## [Editor Report · Decision Letter 2]

20 Aug 2020

Dear Dr Stearns,

On behalf of my colleagues and the Academic Editor, Piali Sengupta, I am pleased to inform you that we will be delighted to publish your Short Reports in PLOS Biology. 

Early Version

PRESS 

Kind regards,

Alice Musson

Publishing Editor, 

PLOS Biology

on behalf of

Roland Roberts,

Senior Editor

PLOS Biology